# Evaluating Post-hoc Explanations for Graph Neural Networks via Robustness Analysis

**Junfeng Fang[1], Wei Liu[1†], Yuan Gao[1], Zemin Liu[2], An Zhang[2],**
**Xiang Wang[1*], Xiangnan He[1*]**
[1]University of Science and Technology of China,
[2]National University of Singapore
{fjf,lw0118,yuangao}@mail.ustc.edu.cn,
{zeminliu,an_zhang}@nus.edu.sg,
{xiangwang1223,xiangnanhe}@gmail.com

## Abstract

This work studies the evaluation of explaining graph neural networks (GNNs), which is crucial to the credibility of post-hoc explainability in practical usage. Conventional evaluation metrics, and even explanation methods — which mainly follow the paradigm of feeding the explanatory subgraph to the model and measuring output difference — mostly suffer from the notorious out-of-distribution (OOD) issue. Hence, in this work, we endeavor to confront this issue by introducing a novel evaluation metric, termed OOD-resistant Adversarial Robustness (OAR). Specifically, we draw inspiration from adversarial robustness and evaluate post-hoc explanation subgraphs by calculating their robustness under attack. On top of that, an elaborate OOD reweighting block is inserted into the pipeline to confine the evaluation process to the original data distribution. For applications involving large datasets, we further devise a Simplified version of OAR (SimOAR), which achieves a significant improvement in computational efficiency at the cost of a small amount of performance. Extensive empirical studies validate the effectiveness of our OAR and SimOAR. Code is available at https://github.com/MangoKiller/SimOAR_OAR.

## 1 Introduction

Post-hoc explainability has manifested its extraordinary power to explain graph neural networks (GNNs) [1, 2, 3, 4]. Given a GNN-generated prediction for a graph, it aims to identify an explanatory subgraph, which is expected to best support the prediction and make the decision-making process more credible, fair, and understandable [5, 6, 7]. However, the reliable evaluation of explanation quality remains a key challenge. As a primary solution, *Human supervision* seeks to justify whether the explanations align with human knowledge [8, 9], but it is often too subjective, thus hardly providing quantifiable assessments. Another straightforward solution is quantitatively measuring the agreement between the generated and ground-truth explanations, such as *Precision* and *Recall* [10, 11]. Unfortunately, access to the ground truth is usually unavailable and labor-extensive, thereby limiting the scope of evaluations based on this method.

Recently, a compromised paradigm — *Feature Removal* [12, 13] — has been prevailing to quantitatively evaluate the explanation's predictive power as compared to the full graph, without exploiting the human supervision and ground truth. The basic idea is to first remove the unimportant features and feed the remaining part (*i.e.,* explanatory subgraph) into the GNN, and then observe how the prediction changes. The prediction discrepancy instantiates *Accuracy* [8] and *Fidelity* [9] of the

---

†Liu Wei is equal contribution to this paper.
*Corresponding authors. Xiang Wang and Xiangnan He are also affiliated with Institute of Artificial Intelligence, Institute of Dataspace, Hefei Comprehensive National Science Center.

37th Conference on Neural Information Processing Systems (NeurIPS 2023).

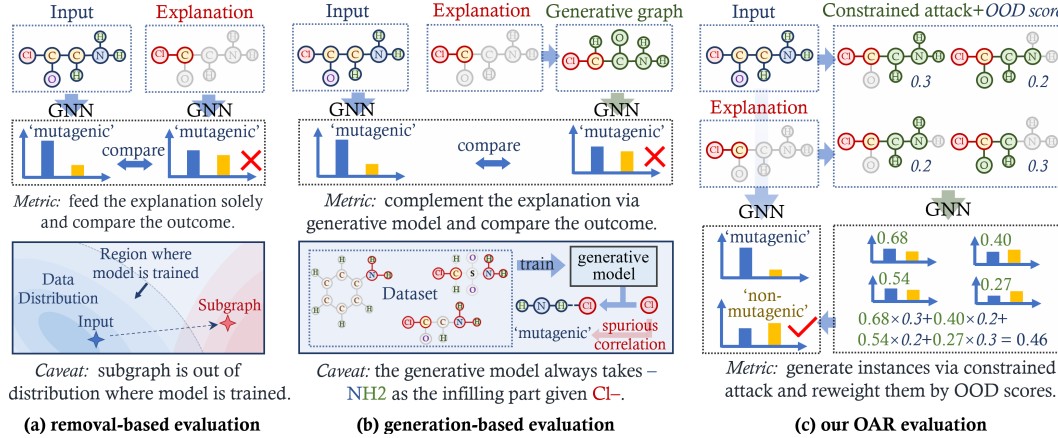

Figure 1: Pipelines and flaws of different evaluation methods. In the "Input" graph, $-NH_2$ is considered as the ground truth explanation for its mutagenicity. Best viewed in color.

explanation, reflecting "how accurate and faithful the explanation is to recover the prediction of the input graph". Despite the prevalence, these removal-based metrics usually come with the caveat of the out-of-distribution (OOD) issue [14, 13]. Specifically, as the after-removal subgraphs are likely to lie off the distribution of full graphs [15, 16], the GNN is forced to handle these off-manifold inputs and easily gets erroneous predictions [17, 18]. Take Figure 1 (a) as an example. For the full molecular graph, the GNN classifies it as "mutagenic", which is reasonable due to the presence of mutagenic $-NH_2$ group; whereas, when taking the subgraph, *i.e.,* non-mutagenic C-Cl group solely as the input, the GNN surprisingly maintains its output "mutagenic". Clearly, the prediction on the explanatory subgraph might be skeptical, which easily deteriorates the faithfulness of the removal-based evaluations.

In sight of this, recent efforts [18, 17] are beginning to mitigate the OOD issue via the *Generation-based* metrics. Instead of directly feeding the to-be-evaluated subgraph into the target GNN, they use a generative model [19, 20] to imagine and generate a new full graph conditioned on the subgraph. These methods believe that the generation process could infill the subgraph and pull it closer to the original graph distribution. As Figure 1 (b) shows, comparing the predictions on this new graph and the original graph could be the surrogate evaluation. While intuitively appealing, the generative models easily inherit the data bias and inject it into the infilling part. Considering Figure 1 (b) again, since molecules in the Mutagenicity dataset comprising non-mutagenic chloride ($-Cl$) always carry amino ($-NH_2$), a generative model is prone to capture this occurrence bias and tend to infill $-NH_2$ with the $-Cl$-involved subgraphs. This bias not only exerts on the generations but also makes the evaluation inconsistent with the GNN behavior: in the generative model, $-Cl$ is assigned with more "mutagenic" scores as it usually accompanies the mutagenic partner $-NH_2$; in contrast, the GNN finds no "mutagenic" clues in $-Cl$. In a nutshell, the generation-based metrics show respect to the data distribution somehow but could be inconsistent with GNNs' behavior and lose control of the infilling part.

Scrutinizing these removal- and generation-based metrics (as summarized in Figure 1), we naturally raise a question: "*Can we design a metric that respects the data distribution and GNN behavior simultaneously?*" To this end, we draw inspiration from adversarial robustness [21, 22] and propose a new evaluation framework, OAR (OOD-resistant Adversarial Robustness), to help reliably assess the explanations. As shown in Figure 1 (c), OAR encapsulates two components: constrained attack and OOD reweighting, which respect the GNN behavior and data distribution, respectively. Specifically:

- Intuitively, perturbations on label-irrelevant features should be ineffective to the GNN prediction, while those on label-relevant features are supposed to be impactful and destructive to the prediction [22, 21]. Hence, for the original input graph, the attack model performs perturbations constrained on the complementary part of its explanation. This perturbing game aims to naturally take control of the "infilling" process, making the explanatory subgraph less influenced by its infilling.
- Having obtained a set of perturbed graphs, the reweighting component estimates the OOD score of each perturbed graph, which reflects the degree of distribution shift from the original data

distribution. Then, we feed these graphs into GNN and reweight their predictions with OOD scores. The sum of weighted predictions can quantitatively evaluate the importance of the target subgraph.

We validate the effectiveness of OAR in evaluation tasks across various start-of-the-art explanation methods, datasets, and backbone GNN models. OAR has manifested surprising consistency with the metrics like *Precision*, *Recall* and *Human Supervision*. Furthermore, to better generalize to the large datasets, we also provide a Simplified version of OAR (SimOAR) achieving significant improvements in computational efficiency at the expense of a small amount of performance degradation. Our main contributions can be summarized as follows:

- We propose a novel metric, OAR for evaluating GNNs' explainability, which tries to resolve the limitation of current removal- and generative-based evaluations by taking both data distribution and GNN behavior into account (Section 2.2).
- We provide a simplified version of OAR, SimOAR for better generalization to the evaluation tasks involving large datasets, which greatly shortens the execution time while only sacrificing a small amount of performance (Section 2.3).
- Experimental results demonstrate that our OAR/SimOAR outperforms the current evaluation metrics by a large margin, and further validate the high efficiency of SimOAR (Section 3).

## 2 Methodology

In this section, we propose an evaluation method for the explainability of GNNs from the perspective of adversarial robustness. We start with the notation of GNNs and its explainability in Section 2.1. After that, we detail our evaluation metric, OAR via three progressive steps (Section 2.2). In Section 2.3, we provide a simplified version of OAR called SimOAR for applications demanding more efficient execution.

### 2.1 Problem Formulation

**Graph neural networks (GNNs).** GNNs have achieved remarkable success due to their powerful representation ability. Without loss of generality, we focus on the graph classification task in this work: a well-trained GNN model $f$ takes a graph $\mathcal{G}$ as the input and outputs its probabilities $\boldsymbol{y}$ over classes $\{1, ..., C\}$, *i.e.*, $\boldsymbol{y} = f(\mathcal{G}) \in \mathbb{R}^C$. Typically, $\mathcal{G}$ is an undirected graph involving the node set $\mathcal{V}$ and the edge set $\mathcal{E}$. We first introduce the feature of node $v_i \in \mathcal{V}$ as a $d$-dimensional vector and collect the features of all nodes into $\boldsymbol{X} \in \mathbb{R}^{|\mathcal{V}| \times d}$. Then we define an adjacency matrix $\boldsymbol{A} \in \mathbb{R}^{|\mathcal{V}| \times |\mathcal{V}|}$ to describe graph topology, where $A_{uv} = 1$ if the edge connecting nodes $u$ and $v$ exists, *i.e.*, $(u, v) \in \mathcal{E}$, otherwise $A_{uv} = 0$. Based on these, $\mathcal{G}$ can be alternatively represented as $\mathcal{G} = (\boldsymbol{A}, \boldsymbol{X})$.

**Explainability for GNNs.** Upon the GNN model $f$, explanation techniques of GNNs generally study the underlying relationships between their outputs $\boldsymbol{y}$ and inputs $\mathcal{G}$. They focus on explainability *w.r.t.* input features, aiming to answer "*Which parts of the input graph contribute most to the model prediction?*". Towards this end, explainers typically assign an importance score to each input feature (*i.e.*, node $v_i$ or edge $(v_i, v_j)$) to trace their contributions. Then they select the salient part (*e.g.*, a subset of nodes or edges with top contributions) as the explanatory subgraph $\mathcal{G}_s$ and delete the complementary part $\mathcal{G}_{\bar{s}} = \mathcal{G} \backslash \mathcal{G}_s$. We formulate the explanation method as $h$ and yield the above process as $\mathcal{G}_s = h(\mathcal{G}, f)$.

### 2.2 OOD-Resistant Adversarial Robustness

Retrospecting the removal- and generation-based evaluations, we emphasize that both these classes come with inherent limitations. Specifically, Removal-based metrics pay less heed to the data distribution thus forcing GNNs to handle off-manifold instances, while generation-based metrics are inconsistent with GNN behavior and lose control of the infilling part. Fortunately, in this section, we claim that it is possible to get the best of and avoid the pitfalls of both worlds — removal-based and generation-based metrics — by taking both GNN behavior and data distribution into account.

To meet these challenges, we elaborate our evaluation metric, OOD-resistant adversarial robustness (OAR) via three progressive steps: **in the first step**, we formulate the adversarial robustness tailored for GNNs explanations, which naturally conforms to the GNN behavior; **in the second step**, we introduce a tractable and easy-to-implement objective of above adversarial robustness; **in the third step**, we introduce an elaborate OOD reweighting block which confines the overall evaluation process to the original data distribution.

**STEP 1: Formulation of Adversarial Robustness.** We prioritize the introduction of *adversarial robustness* in machine learning [21, 22, 23] that motivates our method. Concretely, given a machine

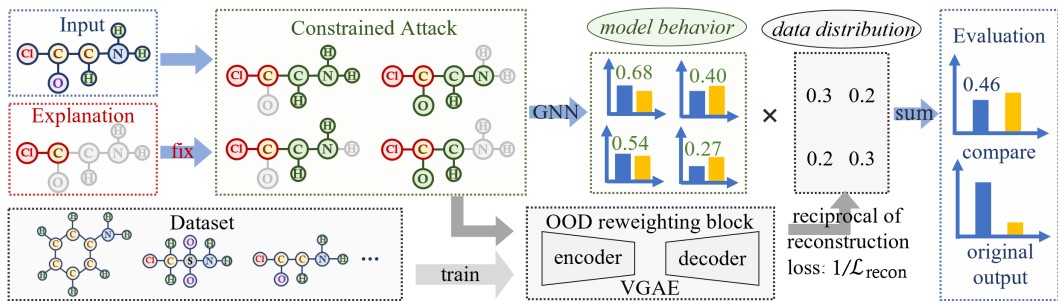

Figure 2: The pipeline of OAR, which takes both model behavior and data distribution into account.

learning model, an input $x$ and a subset of the input $x_s \subseteq x$, the *adversarial robustness* of $x_s$ denotes the minimum perturbation leading to the wrong prediction, on the condition that perturbation is only imposed on $x_s$ [22]. Inspired by this idea, we define the adversarial robustness of GNN explanation $\mathcal{G}_s$, and formulate it as the minimum adversarial perturbation $\delta$ on the structure of complementary subgraph $\mathcal{G}_{\bar{s}}$. More formally,

**Definition 1.** *Given a GNN model $f$, an input graph $\mathcal{G} = (A, X)$ with prediction $y$ and explanation $\mathcal{G}_s$, suppose that $\mathcal{G}' = (A', X')$ is the graph generated by adding and deleting edges in $\mathcal{G}$, the adversarial robustness $\delta$ of explanation $\mathcal{G}_s$ is defined as:*

$$\delta_{\mathcal{G}_s} = \min_{A'} \sum_{u \in \mathcal{V}} \sum_{v \in \mathcal{V} \setminus u} |A_{uv} - A'_{uv}|$$
$$s.t. \ \arg\max_i f(\mathcal{G}')_i \neq \arg\max_i y_i, \quad \sum_{u \in \mathcal{V}_s} \sum_{v \in \mathcal{V}_s \setminus u} |A_{uv} - A'_{uv}| = 0, \tag{1}$$

*where $\mathcal{V}$ and $\mathcal{V}_s$ are the node sets of $\mathcal{G}$ and $\mathcal{G}_s$, respectively.*

*Definition 1* identifies the quality of explanation $\mathcal{G}_s$ as the difficulty of reversing the prediction by perturbing features not belonging to $\mathcal{G}_s$ solely. That is, when $\mathcal{G}_s$ is fixed, the more difficult it is to fool the model by perturbing its complementary, the more important $\mathcal{G}_s$ is. The key intuition behind this inference is: if an explanation comprises most of the label-relevant features, it is conceivably hard to change the prediction by manipulating the remaining features that are label-irrelevant. Thus, according to *Definition 1*, we can find that: a good explanation would yield high adversarial robustness $\delta$ and vice versa, which naturally conforms to the GNN behavior.

It seems that adversarial robustness is the feasible metric to evaluate the GNNs' explanations. However, there are still two matters standing in the way of its adoption: 1) Is its objective (*i.e.*, Equation (1)) tractable and easy to implement? 2) Does it respect the data distribution?

**STEP 2: Finding a Tractable Objective.** To answer the first question, we argue that Equation (1) is hard to realize and sometimes even intractable owing to two possible reasons:

- The primary reason is that adversarial attacks may fail to find any adversarial example, since a solution satisfying two conditions in Equation (1) simultaneously may not exist. In other words, if the explanation $\mathcal{G}_s$ is precise enough, it is almost impossible to reverse the prediction via manipulating features in the complementary part $\mathcal{G}_{\bar{s}}$ which are mainly label-irrelevant.
- It is notoriously hard to search for the minimum adversarial perturbation $\delta$ in most cases. Current attack methods [24, 25] typically turn to find an alternative sub-optimal solution. Thus, leveraging these methods could introduce additional bias and threaten the fairness of evaluation.

To address these issues and make the evaluation objective tractable and easy to implement, we first formulate the inference derived from *Definition 1*:

**Proposition 1.** *When the high quality explanation $\mathcal{G}_s$ is anchored (fixed), perturbations restricted to the complementary part $\mathcal{G}_{\bar{s}}$ have a weak influence on the model prediction.*

While *Definition 1* evaluates the explanations via the adversarial robustness $\delta$, *Proposition 1* indicates a more straightforward way to the tractable evaluation objective from its dual perspective. To be more specific, *Definition 1* quantifies perturbation on $\mathcal{G}_{\bar{s}}$ causing change of prediction; conversely, *Proposition 1* quantifies change of prediction caused by perturbation on $\mathcal{G}_{\bar{s}}$. More formally,

**Definition 2.** *Given a GNN model $f$, an input graph $\mathcal{G} = (\boldsymbol{A}, \boldsymbol{X})$ with prediction $\boldsymbol{y}$ and explanation $\mathcal{G}_s$, suppose that $\mathcal{G}' = (\boldsymbol{A}', \boldsymbol{X}')$ is the graph generated by adding and deleting edges in $\mathcal{G}$, the approximate adversarial robustness $\delta^*$ of $\mathcal{G}_s$ is defined as:*

$$\delta^*_{\mathcal{G}_s} = \mathbb{E}_{\mathcal{G}'}\left(f(\mathcal{G}')_c - \boldsymbol{y}_c\right)$$
$$s.t. \quad c = \arg\max_i \boldsymbol{y}_i, \quad \sum_{u \in \mathcal{V}_s} \sum_{v \in \mathcal{V}_s \setminus u} |A_{uv} - A'_{uv}| = 0, \tag{2}$$

*where $\mathcal{V}_s$ refers to the node set of $\mathcal{G}_s$; $c$ denotes the predicted class of $\mathcal{G}$ by model $f$; $f(\mathcal{G}')_c$ represents the probability value of $f(\mathcal{G}')$ for the given class $c$.*

Stemmed from *Definition 2*, the evaluation method to quantify the adversarial robustness of GNN explanations is more explicit and computationally convenient. As shown in Figure 2, for the to-be-evaluated subgraph $\mathcal{G}_s$ (*i.e.*, C-Cl in red dotted box), we anchor it and randomly perturb the remain part $\mathcal{G}_{\bar{s}}$ to get graphs $\mathcal{G}'$ (*i.e.*, molecules in green dotted box). Then we compare the expectation of the prediction of $\mathcal{G}'$ with the prediction of the original graph $\mathcal{G}$. If they are close, most features in $\mathcal{G}_{\bar{s}}$ must be label-irrelevant. Hence, we can assign high quality for the explanation $\mathcal{G}_s$.

So far, there is only one question left: how to ensure that the aforementioned evaluation process respects the data distribution?

**STEP 3: OOD Reweighting Block Tailored for GNNs.** Before elaborating our OOD block, we first retrospect that in most scenarios, a tiny perturbation would not induce large distribution shifts along the input space, thanks to the approximate continuity of input features (*e.g.*, image pixels in *computer vision*). Unfortunately, the structural features of GNNs' inputs — adjacency matrix comprising of 0s and 1s — are discrete thus only one perturbation (*e.g.*, adding or deleting an edge) could induce large distribution shifts, and further violate the underlying properties, such as node degree distribution [26], graph size distribution [27] and domain-specific constraints [28].

Thus, it is pivotal to construct an OOD reweighting block for assessing whether the generated graph $\mathcal{G}'$ deviates from the data manifold. This block is expected to assign an "OOD score" — the degree of distribution shift between $\mathcal{G}'$ and original graph $\mathcal{G}$ — to each $\mathcal{G}'$. However, it is non-trivial to quantify the degree of OOD [29]. Inspired by the great success of *graph anomaly detection* [30, 31, 32, 33, 34], we treat the OOD instance as the "anomaly" since it is isolated from the original data distribution, and naturally employ the common usage module of anomaly detection — *variational graph auto-encoder* (VGAE) [19] containing an encoder and a decoder — to instantiate our OOD reweighting block. Additionally, the great success of diffusion generative models has been recognized recently [35, 36], and thus this domain is deferred for future investigation.

Concretely, as shown in Figure 2, the preparation for evaluations is training the VGAE model on the dataset $\mathcal{D}$ where input graph $\mathcal{G}$ is sampled. After that, we can leverage the reconstruction loss to approximate the degree of OOD for any generated instance $\mathcal{G}'$. To be more specific:

- Given $\mathcal{G}' = (\boldsymbol{A}', \boldsymbol{X}')$, the **encoder** first learns a latent matrix $\boldsymbol{Z}$ according to $\boldsymbol{A}'$ and $\boldsymbol{X}'$, where row $\boldsymbol{z}_i$ corresponds to the node $v'_i$ in $\mathcal{G}'$. Note that $\boldsymbol{z}_i$ is assumed to follow the independent normal distributions with expectation $\boldsymbol{\mu}_i$ and variance $\boldsymbol{\sigma}_i^2$. Formally:

$$q(\boldsymbol{Z}|\boldsymbol{A}', \boldsymbol{X}') = \prod_{i=1}^{|\mathcal{V}'|} q(\boldsymbol{z}_i|\boldsymbol{A}', \boldsymbol{X}') = \prod_{i=1}^{|\mathcal{V}'|} \mathcal{N}(\boldsymbol{z}_i \mid \boldsymbol{\mu}_i, \mathrm{diag}(\boldsymbol{\sigma}_i^2)), \tag{3}$$

  where $\boldsymbol{\mu}$ and $\boldsymbol{\sigma}$ are parameterized by two two-layer GCNs [37] called $\mathrm{GCN}_{\boldsymbol{\mu}}$ and $\mathrm{GCN}_{\boldsymbol{\sigma}}$.
- Then, the **decoder** recovers the adjacency matrix $\boldsymbol{A}'$ based on $\boldsymbol{Z}$:

$$p(\boldsymbol{A}'|\boldsymbol{Z}) = \prod_{i=1}^{|\mathcal{V}'|} \prod_{j=1}^{|\mathcal{V}'|} p(A'_{ij} \mid \boldsymbol{z}_i, \boldsymbol{z}_j),$$
$$\text{with } p(A'_{ij} = 1 \mid \boldsymbol{z}_i, \boldsymbol{z}_j) = \sigma(\boldsymbol{z}_i^\top \boldsymbol{z}_j), \tag{4}$$

  where $\sigma(\cdot)$ is the logistic sigmoid function.
- The **OOD score** of $\mathcal{G}'$ is given by the normalized reciprocal of the reconstruction loss $\mathcal{L}_{recon}(\mathcal{G}')$,

$$\mathcal{L}_{recon}(\mathcal{G}') = -\log p(\boldsymbol{A}' \mid \boldsymbol{Z}), \text{ with } \boldsymbol{Z} = \boldsymbol{\mu} = \mathrm{GCN}_{\boldsymbol{\mu}}(\boldsymbol{A}', \boldsymbol{X}'). \tag{5}$$

Since VGAE is trained on the dataset $\mathcal{D}$, $\mathcal{G}'$ straying far from the data distribution of $\mathcal{D}$ would get the high reconstruction loss $\mathcal{L}_{recon}$. Thus, as the reciprocal of $\mathcal{L}_{recon}$, the OOD score of $\mathcal{G}'$ must be low. Conversely, if $\mathcal{G}'$ is in distribution, it would gain a high OOD score because it is easy to be reconstructed. Based on this, our OOD block can mitigate the impact of OOD instances. Specifically, the OOD score is utilized as the weight of each prediction when calculating the expectation of the generated graph's prediction. This allows for the marginalization of instances with low OOD scores, as shown in the gray dotted box of Figure 2.

**Overall evaluation process.** As the last piece of the OAR puzzle, *i.e.,* OOD reweighting block has been instantiated, let's revisit Figure 2 and summarize the overall process of OAR:

1. Before we evaluate the explanatory subgraph $\mathcal{G}_s$, the OOD reweighting block (*i.e.,* VGAE) is trained on the dataset $\mathcal{D}$ where input graph $\mathcal{G}$ is sampled.
2. Then, we fix the $\mathcal{G}_s$ and randomly perturb the structure of the complementary part $\mathcal{G}_{\bar{s}}$ to get $\mathcal{G}'$.
3. Each $\mathcal{G}'$ is fed into GNN $f$ and VGAE simultaneously to audit prediction and OOD score. Both GNN's behavior and data distribution are taken into consideration in this step.
4. At last, according to the predictions and their weights (*i.e.,* OOD scores), we calculate the weighted average of the generated graphs' predictions. The closer this average is to the original prediction of $\mathcal{G}$, the higher the quality of the explanation $\mathcal{G}_s$ is.

The pseudocode and the tricks to expedite computations are detailed in Appendix A.

## 2.3 A simplified version of OAR

To better generalize to large datasets and reduce the computational complexity, we provide a simplified version of OAR called SimOAR in this section. Compared with OAR, SimOAR achieves a significant improvement in computational efficiency at the expense of a small amount of performance degradation. Concretely, SimOAR is mainly motivated by three empirical inferences after executing OAR:

- The most time-consuming part of OAR is its preparatory work, *i.e.,* training OOD reweighting block, especially for large datasets. For example, on the dataset *MNIST superpixels* [38] containing 70,000 graphs, the converged process of VGAE occupies 93.7% of OAR's execution time.
- In the course of generating $\mathcal{G}'$, the number of perturbation operations is roughly proportional to the degree of distribution shift given by the OOD block. For example, the graph $\mathcal{G}'_1$ created via deleting one edge typically gets a higher OOD score than the graph $\mathcal{G}'_2$ created via deleting five edges.
- For two generated graphs generated via the same perturbation times, they generally get similar reconstruction losses and are assigned similar OOD scores.

Based on these, to expedite computations and simplify the OAR, we deactivate the OOD reweighting block (*i.e.*, deleting all the sketches in gray dotted boxes in Figure 2) in OAR. As compensation for data distribution, we restrict the ratio of the number of perturbations to the number of edges in the original graph $\mathcal{G}$ to a pre-defined minor value $R$. Since the generated graphs typically share similar reconstruction losses and OOD scores while $R$ is fixed, we directly calculate their average prediction to approximate their excepted prediction. The pseudocode and more implementation details of SimOAR are provided in Appendix A.

It is worth noting that despite the potential existence of a few generated graphs $\mathcal{G}'$ in SimOAR that fall outside the distribution, the performance of SimOAR still significantly surpasses that of current evaluation methods *w.r.t* consistency with both metrics based on ground truth and human supervision. Hence, in light of the efficiency of the SimOAR, we strongly advocate for its adoption as a predominant alternative to prevalent removal-based evaluation metrics. At the heart of SimOAR – and a **central thesis** of this paper – is the perspective that, during evaluation, rather than deleting all non-explanatory nodes and then gauging the resultant output variations, it is more insightful to **randomly delete a portion of the non-explanatory nodes multiple times and then gauge the average output variations.**

## 3 Experiments

We present empirical results to demonstrate the effectiveness of our proposed methods OAR and SimOAR. The experiments aim to investigate the following research questions:

- **RQ1:** How is the evaluation quality of OAR and SimOAR compared to that of existing metrics?
- **RQ2:** How is the generalization of OAR and SimOAR compared to that of existing metrics?
- **RQ3:** What is the impact of the designs (*e.g.,* the OOD reweighting block) on the evaluations?

Table 1: Overall evaluation scores and rankings of explainers under different evaluation methods. Symbol $(\cdot)$ indicates the rank of explainers. Our methods, *i.e.,* OAR and SimOAR are bold and the best-performing methods are denoted with the superscript asterisk.

| | | SA | GradCAM | GNNExplainer | PGExplainer | CXPlain | ReFine | $\tau \uparrow$ |
|---|---|---|---|---|---|---|---|---|
| BA3 | Recall | $88.12_{\pm0.00}(1)$ | $84.53_{\pm0.00}(2)$ | $76.83_{\pm4.64}(3)$ | $65.32_{\pm5.21}(5)$ | $54.77_{\pm4.42}(6)$ | $72.90_{\pm3.72}(4)$ | - |
| | RM | $35.39_{\pm0.00}(3)$ | $37.67_{\pm0.00}(4)$ | $43.32_{\pm1.97}(1)$ | $29.25_{\pm2.26}(5)$ | $27.78_{\pm0.96}(6)$ | $41.24_{\pm1.55}(2)$ | 0.73 |
| | DSE | $43.08_{\pm2.31}(1)$ | $41.38_{\pm1.75}(2)$ | $22.25_{\pm2.14}(6)$ | $37.92_{\pm2.52}(3)$ | $24.60_{\pm3.57}(5)$ | $29.31_{\pm2.96}(4)$ | 0.73 |
| | **OAR** | $93.12_{\pm4.60}(1)$ | $86.20_{\pm3.76}(2)$ | $80.19_{\pm1.68}(3)$ | $65.48_{\pm3.75}(5)$ | $59.69_{\pm2.98}(6)$ | $71.02_{\pm4.45}(4)$ | **1.00\*** |
| | **SimOAR** | $84.39_{\pm5.70}(1)$ | $83.44_{\pm4.81}(2)$ | $62.52_{\pm2.25}(3)$ | $50.02_{\pm2.87}(6)$ | $55.49_{\pm4.22}(5)$ | $60.42_{\pm3.32}(4)$ | **0.93** |
| TR3 | Recall | $82.08_{\pm0.00}(1)$ | $77.00_{\pm0.00}(2)$ | $60.09_{\pm4.97}(4)$ | $55.85_{\pm4.70}(5)$ | $44.39_{\pm5.57}(6)$ | $74.19_{\pm3.30}(3)$ | - |
| | RM | $55.08_{\pm0.00}(3)$ | $51.15_{\pm0.00}(4)$ | $79.08_{\pm4.31}(1)$ | $50.45_{\pm2.04}(5)$ | $47.72_{\pm3.82}(6)$ | $64.60_{\pm2.27}(2)$ | 0.67 |
| | DSE | $48.51_{\pm1.00}(1)$ | $37.32_{\pm2.35}(4)$ | $44.82_{\pm2.90}(2)$ | $33.71_{\pm3.72}(6)$ | $35.65_{\pm1.92}(5)$ | $39.49_{\pm3.31}(3)$ | 0.73 |
| | **OAR** | $95.23_{\pm3.75}(1)$ | $87.51_{\pm6.65}(2)$ | $72.63_{\pm5.89}(3)$ | $59.06_{\pm4.83}(5)$ | $51.61_{\pm3.14}(6)$ | $63.82_{\pm5.43}(4)$ | **0.93\*** |
| | **SimOAR** | $88.45_{\pm6.04}(1)$ | $76.37_{\pm4.54}(2)$ | $53.54_{\pm4.46}(6)$ | $68.58_{\pm5.80}(4)$ | $62.98_{\pm4.00}(5)$ | $75.59_{\pm4.27}(3)$ | **0.86** |
| MNIST-sp | Recall | $43.98_{\pm0.00}(3)$ | $44.39_{\pm0.00}(4)$ | $54.63_{\pm0.96}(1)$ | $30.13_{\pm1.42}(6)$ | $38.96_{\pm2.62}(5)$ | $47.88_{\pm1.60}(2)$ | - |
| | RM | $21.34_{\pm0.00}(4)$ | $19.10_{\pm0.00}(5)$ | $22.23_{\pm1.02}(3)$ | $25.04_{\pm0.35}(2)$ | $27.15_{\pm0.69}(1)$ | $17.58_{\pm0.50}(6)$ | 0.33 |
| | DSE | $30.37_{\pm4.06}(2)$ | $29.19_{\pm2.19}(3)$ | $14.03_{\pm1.77}(6)$ | $28.45_{\pm2.65}(1)$ | $22.95_{\pm2.44}(4)$ | $21.32_{\pm1.29}(5)$ | 0.20 |
| | **OAR** | $66.28_{\pm2.46}(3)$ | $64.18_{\pm5.25}(4)$ | $82.22_{\pm4.13}(1)$ | $63.88_{\pm3.45}(5)$ | $51.37_{\pm1.76}(6)$ | $75.43_{\pm4.84}(2)$ | **0.93\*** |
| | **SimOAR** | $54.72_{\pm3.84}(4)$ | $69.86_{\pm2.80}(3)$ | $79.69_{\pm3.53}(1)$ | $33.27_{\pm2.04}(6)$ | $52.93_{\pm3.26}(5)$ | $76.40_{\pm2.24}(2)$ | **0.93\*** |
| MUTAG | Recall | $41.12_{\pm0.00}(6)$ | $44.44_{\pm0.00}(5)$ | $55.95_{\pm4.94}(4)$ | $71.22_{\pm2.54}(2)$ | $64.65_{\pm1.50}(3)$ | $77.73_{\pm3.67}(1)$ | - |
| | RM | $75.60_{\pm0.00}(5)$ | $77.39_{\pm0.00}(6)$ | $82.32_{\pm4.37}(2)$ | $87.11_{\pm2.59}(1)$ | $76.49_{\pm3.51}(4)$ | $81.76_{\pm2.64}(3)$ | 0.73 |
| | DSE | $38.03_{\pm3.90}(4)$ | $32.36_{\pm2.40}(5)$ | $40.28_{\pm1.32}(3)$ | $49.18_{\pm2.57}(1)$ | $29.06_{\pm1.44}(6)$ | $43.19_{\pm2.14}(2)$ | 0.67 |
| | **OAR** | $52.10_{\pm3.58}(6)$ | $53.19_{\pm3.87}(5)$ | $63.35_{\pm2.50}(4)$ | $88.46_{\pm2.43}(2)$ | $67.19_{\pm2.08}(3)$ | $92.81_{\pm5.24}(1)$ | **1.00\*** |
| | **SimOAR** | $75.45_{\pm2.51}(5)$ | $71.07_{\pm5.40}(6)$ | $85.65_{\pm4.00}(2)$ | $81.48_{\pm2.98}(3)$ | $76.32_{\pm3.88}(4)$ | $89.40_{\pm5.06}(1)$ | **0.80** |

## 3.1 Experimental settings

To evaluate the effectiveness of our method, we utilize four benchmark datasets: BA3 [39], TR3 [17], Mutagenicity [40, 41], and MNIST-sp [38], which are publicly accessible and vary in terms of domain and size. Moreover, to generate the explanations of the graphs in the datasets mentioned above, we adopt several state-of-the-art post-hoc explanation methods, *i.e.,* SA [10], GradCAM [42], GNNExplainer [11], PGExplainer [43], CXPlain [44] and ReFine [39]. The prevailing metrics — removal-based evaluation (RM for short) and generation-based evaluation, *i.e.,* DSE [17] — are selected as the baselines. Detailed experimental details can be found in Appendix B.

## 3.2 Measurement metric

We elaborate on the measurement metric of existing evaluation methods in this part since how to fairly define the quality of an evaluation method is critical to our research.

**Ground-truth explanations.** We first follow the prior studies [11, 45, 43] and treat the subgraphs coherent to data generation procedure or human knowledge as ground truth. Although ground truth might not conform to the decision-making process exactly, it contains sufficient discriminative information to help justify the quality of explanations. Moreover, it's worth emphasizing again that our method does not depend on ground truth, and gathering ground truth is only for fair comparison.

**Consistency with metric based on ground-truth explanations.** Specifically, given a to-be-evaluated subgraph $\mathcal{G}_s$ and its corresponding ground-truth explanation $\mathcal{G}_s^{GT}$, we use **Recall** as the gold evaluation metric defined as $\text{Recall}(\mathcal{G}_s) = \left| \mathcal{E}_s \bigcap \mathcal{E}_s^{GT} \right| / \left| \mathcal{E}_s^{GT} \right|$, where $\mathcal{E}_s$ and $\mathcal{E}_s^{GT}$ are the edge set of $\mathcal{G}_s$ and $\mathcal{G}_s^{GT}$; $|\cdot|$ denotes the cardinal function of set. Hence, for any evaluation method, we can calculate its consistency with Recall to quantify its performance via Kendall correlation $\tau$ [46] defined as:

$$\tau\left(\{r^i\}_{i=1}^n, \{s^i\}_{i=1}^n\right) = \frac{2}{n(n+1)} \sum_{i<j} I\left(\text{sgn}\left(r^i - r^j\right) = \text{sgn}\left(s^i - s^j\right)\right), \qquad (6)$$

where $\{r^i\}_{i=1}^n$ and $\{s^i\}_{i=1}^n$ are a pair of Recall values and evaluation scores; $\text{sgn}(\cdot)$ is the sign function and $I(\cdot)$ is the indicator function. The bigger $\tau$ is, the higher the evaluation scores are consistent with Recall values, and thus the better the evaluation method is.

**Consistency with human intuition.** The consistency between evaluation results and human intuition is also an important reference. In view of the high subjectivity of human intuition, we organized a large-scale user study engaging 100 volunteers. Results are shown in Appendix C.

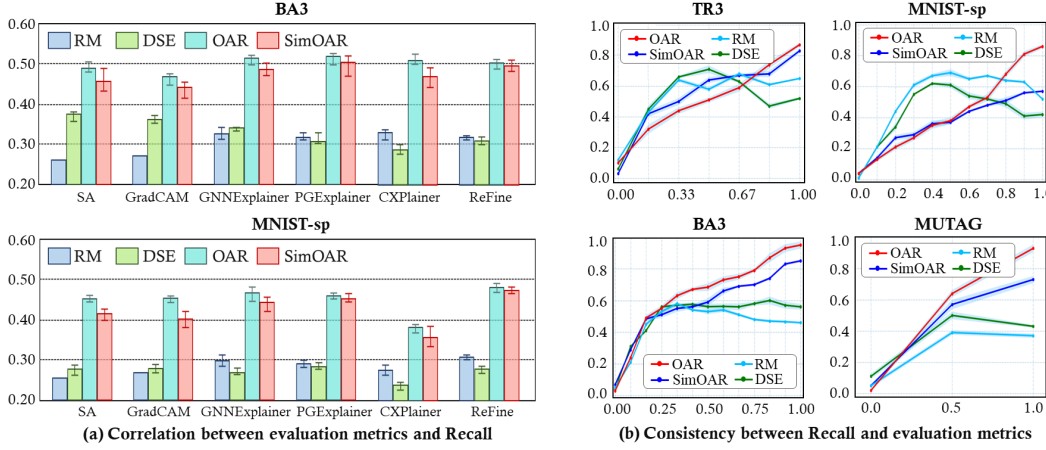

Figure 3: The performance of various evaluation metrics. (a) Correlation between metrics and Recall across various backbone explainers, where the vertical axis represents the normalized Kendall rank correlation. (b) Consistency between Recall and the scores provided by metrics. The more monotonously increasing the curve is, the better the evaluation metric is. Best viewed in color.

Limited by space, we further exhibit and discuss more detailed implementations and results in Appendix B, including but not limited to the generalization of OAR involving the evaluation of the post-hoc explanations for node classification task and the inherent explanations.

### 3.3 Study of explanation evaluation (RQ1)

As a preparation for the experiments, we first collect the explanatory subgraphs $\{\mathcal{G}_s^i\}$ for a set of graphs $\{\mathcal{G}^i\}_{i=1}^N$ and the corresponding well-trained GNN model $f$. We denote the evaluation score on $\mathcal{G}_s^i$ based on RM, DSE and our OAR and SimOAR by $s_{\mathrm{RM}}^i$, $s_{\mathrm{DSE}}^i$, $s_{\mathrm{OAR}}^i$ and $s_{\mathrm{SimOAR}}^i$, respectively. For more faithful comparison, we present both the explainer-level correlation defined as $\tau_* = \tau\left(\left\{\frac{1}{N}\sum_{i=1}^N \mathrm{Recall}\left(\mathcal{G}_s^{i,h}\right)\right\}_{h\in\mathcal{H}}, \left\{\frac{1}{N}\sum_{i=1}^N s_*^{i,h}\right\}_{h\in\mathcal{H}}\right)$ and the instance-level correlation defined as $\tau_* = \tau\left(\left\{\mathrm{Recall}\left(\mathcal{G}_s^i\right)\right\}_{i=1}^N, \left\{s_*^i\right\}_{i=1}^N\right)$, where $*$ can be RM, DSE, OAR, and SimOAR; $\mathcal{G}_s^{i,h}$ means the subgraph $\mathcal{G}_s^i$ is extracted by explainer $h$, $\mathcal{H}$ is the set of explainers. The explainer-level results, under all evaluation methods, on all datasets, are presented in Table 1. Moreover, considering the instance-level results share a similar tendency, we presented the representative results on BA3 and MNIST-sp in Figure 3 (a). According to Table 1 and Figure 3 (a) we can find that:

- **Observation 1: OAR outperforms other methods in all cases.** Substantially, Kendall rank correlation greatly improves after leveraging the paradigm of adversarial robustness. The most notable case is the explainers' rank on BA3 and MUTAG, where $\tau^* = 1.00$ achieves a tremendous increase from the RM and the DSE. It demonstrates the effectiveness and universality of OAR/SimOAR and verifies that OAR/SimOAR can be leveraged to boost the quality of evaluations.
- **Observation 2: SimOAR performs a little worse than OAR, but still significantly improves over the strongest baselines.** To be more specific, the average score of SimOAR is 7.83% less than that of OAR, but still, 42.62% higher than RM and 37.45% higher than DSE. It demonstrates that SimOAR is adequate for the task of explanation evaluations in most cases.

Further analysis of the results presented in Table 1 reveals that:

- **Observation 3: OAR/SimOAR presents a more fair and faithful comparison among explainers.** The rankings provided by the OAR and SimOAR are highly consistent (*i.e.*, $\tau^* = 0.928$ on average) with the references, while the removal- and generation-based rankings unsurprisingly pale by comparison. These empirical results give us the courage to leverage OAR and SimOAR to evaluate emerging explanation methods in the future.

### 3.4 Study of generalization (RQ2)

Although the experiment provided in 3.3 is detailed and fair, we contend that the generalization of OAR and SimOAR is still unexamined. Specifically, a specific explainer always has a preference

for certain patterns, and thus the explanatory subgraphs extracted by it often have similar structures. Therefore, the experimental results based on limited explainers may not generalize well to other existing or future explainers, especially those based on different lines of thought.

As it is impractical for us to cover all these explainers, we resort to directly generalizing the to-be-evaluated subgraphs. That is, we make use of fake explanatory subgraphs which are randomly sampled from the full graph. The detailed sampling algorithm and settings can be found in Appendix C. In these settings, the best case is that the evaluation score is monotonically increasing *w.r.t.* the Recall level, which indeed reaches the best consistency. Average normalized scores under all evaluation methods are shown in Figure 3 (b), which indicate that:

- **Observation 4: OAR/SimOAR greatly improves the consistency between evaluation scores and Recalls, which indicates that our method has tremendous potential to perform well on other explainers.** Conversely, removal- and generation-based methods are negatively correlated with the importance involving the set of to-be-evaluated subgraphs which get high Recalls.

### 3.5 Study of designs (RQ3)

**Effectiveness of OOD block.** We first focus on the effectiveness of the OOD block. The most immediate impact of OOD block on OAR can be estimated by comparing the performance of OAR and SimOAR, which can be also deemed as the ablation experiments. Hence, we turn to qualitatively analyze the OOD block via some case studies shown in Figure 4, where all the OOD scores belonging to the same datasets are normalized to the range of 0 to 1. Based on the information conveyed in Figure 4, the following observation can be made:

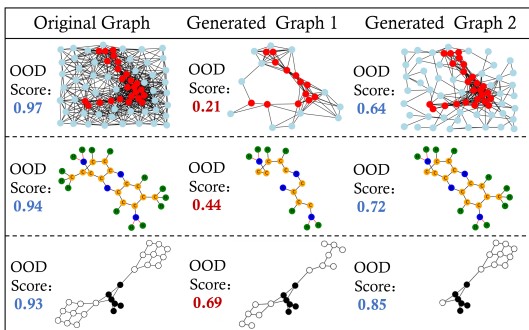

Figure 4: Case studies for OOD reweighting block with the graphs randomly selected from datasets MNIST-sp, MUTAG, and BA3 arranged from top to bottom. Best viewed in color.

- **Observation 5: OOD block can assign the lower weights to the subgraph which violent the underlying properties of the full graph.** For example, graph properties of chemical molecules, such as the valency rules, impose some constraints on syntactically valid molecules. Hence, the invalid molecular subgraphs, which destroy the integrity of the carbon ring by simply removing some bonds (edges) or atoms (nodes) are assigned low scores by the OOD block.

**Time complexity.** To further explore the efficiency of our evaluation method and the designed module in it, we count the running time of the evaluation process on every single graph and average the time over the entire test set to obtain per-graph time consumption. The comparison is provided in Table 2. According to Table 2 we can find that:

Table 2: Per-graph time consumption. (ms)

|        | BA3           | TR3           | MNIST-sp      | MUTAG         |
|--------|---------------|---------------|---------------|---------------|
| RM     | $0.11_{\pm 0.02}$ | $0.10_{\pm 0.01}$ | $0.15_{\pm 0.03}$ | $0.13_{\pm 0.01}$ |
| DSE    | $1.73_{\pm 0.27}$ | $1.60_{\pm 0.14}$ | $2.12_{\pm 0.39}$ | $1.88_{\pm 0.25}$ |
| OAR    | $1.16_{\pm 0.05}$ | $1.32_{\pm 0.06}$ | $1.77_{\pm 0.10}$ | $1.78_{\pm 0.03}$ |
| SimOAR | $0.84_{\pm 0.07}$ | $0.65_{\pm 0.08}$ | $1.03_{\pm 0.07}$ | $0.96_{\pm 0.06}$ |

- **Observation 6: SimOAR greatly reduces the execution time.** Concretely, the execution speed has nearly doubled after leveraging the metric of SimOAR. This significant improvement in efficiency corresponds to the original intention of SimOAR and verifies the success of SimOAR's design.

## 4 Related Work & Further Discussion

**OAR & Contemporary Evaluation Metrics.** Apart from the qualitative evaluation methods based on human intuition, recent literature categorizes quantitative metrics into four primary categories: accuracy, faithfulness, stability, and fairness [47, 48, 49, 50, 51, 52]. Notably, Precision and Recall metrics align with the accuracy category, while our proposed OAR and SimOAR fall under the faithfulness category. Among these methods, the most recently proposed faithfulness metric is GEF [49], which however omits quantification of the distribution shift in subgraphs. Nevertheless, we

have exhibited the experimental comparison between GEF and our metrics *w.r.t* the latest dataset SHAPEGGEN [49] in Appendix B.

**OOD & GNNs Explainability** The OOD issue is one of the most critical challenges in the post-hoc explainability domain currently [16, 53]. To sidestep this challenge, many studies have pivoted towards the development of inherently explainable GNNs [15, 54]. Notwithstanding the complexity of the task, efforts such as FIDO [55] are making significant successes in addressing the OOD problem within post-hoc explanations. Concurrently, CGE [56] leverages the lottery ticket hypothesis [57, 58, 59, 60, 61] to craft the cooperative explanation for both GNNs and graphs, wherein the OOD challenge is potentially mitigated through the EM algorithm. GIBE [62] delves into the intersection of the OOD issue and regularization, viewing it through the lens of information theory. Furthermore, MixupExplainer [63] and CoGE [64] navigate the OOD problem from the generation and recognition stances, respectively.

Simultaneously, the evaluation of inherent explanations encounters the same hurdles as post-hoc explanations: it's challenging to quantify in the absence of the ground truth. Fortunately, by introducing an additionally well-trained GNN, OAR can be employed to **evaluate inherent explanations** in a similar way to evaluate post-hoc explanations. The experimental results are shown in Appendix B.

**Limitations & Concerns.** While we acknowledge the effectiveness of our methods, we also recognize their limitations. Concretely, despite utilizing SimOAR to expedite the evaluation process, our paradigm remains more time-intensive compared to the conventional removal-based metric. To overcome this constraint, a probable solution is summarizing the optimal number of perturbations and implementing a self-adaptive extraction module to select the perturbed features.

Furthermore, we recognize potential apprehensions regarding the migration of trust issues from the black-box GNN to the equally non-transparent VGAE. Nevertheless, we posit that the repercussions of this are substantially mitigated in our streamlined method. Specifically, SimOAR bypasses VGAE in favor of employing transparent heuristics for perturbation generation, effectively addressing the aforementioned trust concerns. It's noteworthy that, while SimOAR's performance may be marginally below or comparable to OAR's, it consistently exceeds other benchmarks. This emphasizes VGAE's restrained impact and reaffirms our recommendation of SimOAR over OAR.

## 5 Conclusion

In this paper, we explored the evaluation process of GNN explanations and proposed a novel evaluation metric, OOD-resistant adversarial robustness (OAR). OAR gets inspiration from the notion of adversarial robustness and evaluates the quality of explanations by calculating their robustness under attack. It addresses the inherent limitations of current removal- and generation-based evaluation metrics by taking both data distribution and GNN behavior into account. For applications involving large datasets, we introduce a simplified version of OAR (SimOAR), which achieves a significant increase in computational efficiency at the cost of a small amount of performance degradation. This work represents an initial attempt to exploit evaluation metrics for post-hoc GNN explainability from the perspective of adversarial robustness and resistance to OOD.

## 6 Ethics Statement

This work is primarily foundational in GNN explainability, focusing on the development of a more reliable evaluation algorithm. Its primary aim is to contribute to the academic community by enhancing the understanding and implementation of the evaluation process. We do not foresee any direct, immediate, or negative societal impacts stemming from the outcomes of our research.

## Acknowledgments

This work was supported by the National Natural Science Foundation of China (9227010114, 62121002) and the University Synergy Innovation Program of Anhui Province (GXXT-2022-040).

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

# A  Algorithms

Algorithm 1 presents the pseudocode of the evaluation process of our proposed method OAR. The pseudocode of SimOAR can be obtained by removing the line 1, 6, and 8 and simply modifying line 9 into "$s \leftarrow \frac{1}{N_{adv}} \sum_i y^{(i)}$". For clarity, we put the step of feeding adversarial graph $\mathcal{G}'^{(i)}$ into the target GNN and the VGAE under the for-loop. However, when implementing in real code, we can batch all those $N_{adv}$ adversarial graphs and feed them at one time, after the sampling process is finished, to expedite computation. Meanwhile, Algorithm 2 presents the sampling process of fake explanatory subgraphs for general evaluation.

---

**Algorithm 1** Evaluation Process of OAR

---

**Input:** Trained GNN $f$; To-be-evaluated subgraph $\mathcal{G}_s$ and its corresponding original graph $\mathcal{G}$ and dataset $\mathcal{D}$; Perturbation ratio $R$; Number of adversarial graphs $N_{adv}$
**Output:** Evaluation score $s$
 1: Train a standard VGAE on $\mathcal{D}$ according to [19].
 2: $c \leftarrow \arg\max_i f(\mathcal{G})_i$.
 3: **for** $i = 1, 2, \ldots, N_{adv}$ **do**
 4:   $\mathcal{G}'^{(i)} \leftarrow$ randomly deleting $\lfloor R \cdot |\mathcal{E}_{\mathcal{G}}| \rfloor$ edges from $\mathcal{G}$ while fixing $\mathcal{G}_s$.
 5:   $y^{(i)} \leftarrow f(\mathcal{G}'^{(i)})_c$.
 6:   $\mathcal{L}_{recon}^{(i)} \leftarrow$ calculated according to Equation 5 using the trained VGAE.
 7: **end for**
 8: $w_{OOD}^{(i)} \leftarrow \frac{1/\mathcal{L}_{recon}^{(i)}}{\sum_j 1/\mathcal{L}_{recon}^{(j)}}, i = 1, 2, \ldots, N_{adv}$.
 9: $s \leftarrow \sum_i w_{OOD}^{(i)} \cdot y^{(i)}$.

---

**Algorithm 2** Sampling Fake Explanatory Subgraphs for General Evaluation

---

**Input:** Dataset $\mathcal{D} = \{\mathcal{G}^1, \mathcal{G}^2, \ldots, \mathcal{G}^N\}$; Number of Recall levels $N_L$; Number of sampled subgraphs per graph $N_{sub}$; Size of sampled subgraph $K_{sub}$
**Output:** Pairs of Recall level and corresponding subgraphs $(L_k, \{G_{s,k}^{i,j} \mid i = 1, 2, \ldots, N; j = 1, 2, \ldots, N_{sub}\}), k = 1, 2 \ldots, N_L$
 1: **for** $k = 1, 2, \ldots, N_L$ **do**
 2:   $L_k \leftarrow \frac{k-1}{N_L - 1}$
 3:   **for** $i = 1, 2, \ldots, N$ **do**
 4:     $K^{GT} \leftarrow$ the number of edges in the ground-truth explanation of $\mathcal{G}^i$
 5:     $K_{pos} \leftarrow \lfloor L_k \times K^{GT} \rceil$
 6:     $K_{neg} \leftarrow K_{sub} - K_{pos}$
 7:     **for** $j = 1, 2, \ldots, N_{sub}$ **do**
 8:       $\mathcal{G}_{s,k}^{i,j} \leftarrow$ a connected subgraph, randomly sampled from $\mathcal{G}$, with $K_{pos}$ edges in the ground truth explanation and $K_{neg}$ edges not in it.
 9:     **end for**
10:   **end for**
11: **end for**

---

# B  Experimentals

## B.1  Experimental Details

All experiments are conducted on a Linux machine with 8 NVIDIA GeForce RTX 3090 (24 GB) GPUs. CUDA version is 11.6 and Driver Version is 510.39.01. All codes are written under Python 3.9.13 with PyTorch 1.13.0 and PyTorch Geometric (PyG)[65] 2.2.0. We adopt the Adam optimizer throughout all experiments.

Overall, for each dataset, a target GNN classification model is well-trained first. Then the explainers are built on the GNN and generate explanations for its prediction on the dataset. After that, the explanation evaluation methods evaluate how well the explanations are. Our work stands at the last level.

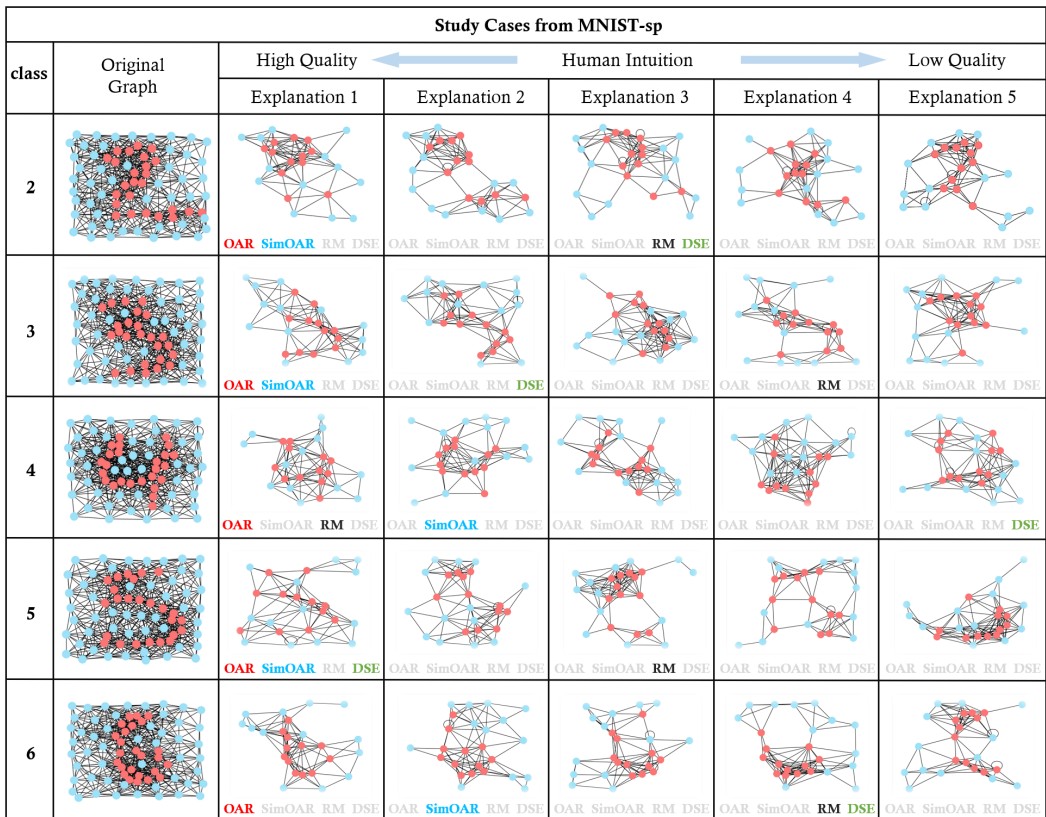

Figure 5: Study cases. For each row, the explanations are ranked based on average rankings given by volunteers. Highlighted evaluation method below each explanation means that the method has given the explanation the highest score compared to other explanations in that row. Best viewed in color.

**Target GNNs.** GNNs have garnered significant recognition for their prowess in encoding graph data [37, 66, 67, 34, 58]. Amidst the vast landscape of GNNs, GIN [67] distinguishes itself with its superior encoding aptitude. In sight of this, the target GNNs for BA3, TR3, and Mutagenicity have the same structure, which is a two-layered GIN followed by a two-layered MLP with 32 hidden channels. They are trained with max epochs equal to 20, 200, and 200 respectively, batch size equal to 128, and learning rate equal to 0.001. The target GNN for MNIST-sp is adapted from an example code[1] provided by PyG, trained with the number of epochs equal to 20, batch size equal to 64, and initial learning rate equal to 0.01. Before training, we randomly split BA3, TR3, and Mutagenicity into train and test sets with ratios of 90% and 10%, respectively, while adopting the split provided by PyG for MNIST-sp. During training, we reserve data of the same size as the test set from the train set as the validation set and save the model which reaches the highest classification accuracy on the validation set for later use.

**Explainers.** We have implemented six state-of-the-art post-hoc explainers, namely, SA, GradCAM, GNNExplainer, PGExplainer, CXPlain, and ReFine, as claimed in Section 3.1, to generate explanatory subgraphs. Here we give a brief introduction to them:

- **SA** [10] captures the gradients *w.r.t.* adjacency matrix of the input features in the process of backpropagation and directly treats them as the importance scores.
- **GradCAM** [42] takes one step further over SA via improving the gradients *w.r.t.* the input features like edges by using context within the graph convolutional layers.
- **GNNExplainer** [11] directly learns an adjacency matrix mask by maximizing the mutual information between a GNN's prediction and distribution of possible subgraph structures.
- **PGExplainer** [43] adopts a deep neural network to parameterize the generation process of explanations, which makes it a natural approach to explaining multiple instances collectively.

---

[1]Reference code for the target GNN of MNIST-sp: `https://github.com/pyg-team/pytorch_geometric/blob/89a54d9454d3832f814f9a574ed421c58f1fce10/examples/mnist_voxel_grid.py`

Table 3: Average consistency scores of different evaluation metrics (corresponding to Table 1 in the main paper).

|  | SHAPEGGEN | TR3 | MUTAG | MNIST | BA3 |
|---|---|---|---|---|---|
| GEF | 0.800 | 0.734 | 0.800 | 0.934 | 0.734 |
| SimOAR | 0.800 | 0.867 | 0.800 | 0.934 | 0.934 |
| OAR | 0.867 | 0.934 | 1.000 | 0.934 | 1.000 |

Table 4: Correlation between metrics and Recall across explanatory subgraphs in four node classification datasets in [11] (corresponding to Figure 3 (a) in the main paper).

|  | BA-Shapes | BA-Community | Tree-Cycles | Tree-Grid |
|---|---|---|---|---|
| RM | 0.312 | 0.321 | 0.295 | 0.411 |
| DSE | 0.406 | 0.377 | 0.343 | 0.384 |
| OAR | **0.542** | **0.560** | **0.489** | **0.440** |
| SimOAR | 0.527 | 0.535 | 0.471 | 0.431 |

- **CXPlain** [44] treats the explanations as a causal learning task and trains causal explanation models that learn to estimate to what degree certain inputs cause outputs in the to-be-explained model.
- **ReFine** [39] leverages the pre-training explanations to exhibit global explanations and the fine-tuning explanations to adapt the global explanations in the local context.

The default hyper-parameters suggested by those papers are adopted, as whether the explainers are at their optimal state is secondary to our work. Among these explainers, SA, GradCAM, and GNNExplainer directly take the to-be-explained graph as input, while the rest three need to be trained on a set of graphs, *i.e.,* the train set in our case, in advance. We only use the explanations extracted from graphs in the test set for evaluation.

**Evaluation Methods.** Finally, we arrive at the explanation evaluation level. There are four evaluation methods, *i.e.,* removal-based evaluation, DSE, OAR, and SimOAR, to be considered. Removal-based evaluation directly feeds the explanatory subgraph into the target GNN and gets its prediction, on the predicted class of the original graph, as the evaluation score, which does not involve any details. For DSE, we make use of its public source code[2] and follow its paper to set hyper-parameters. As for our method OAR/SimOAR, we have summarized the entire process of OAR in Algorithm 1. Here we present more details on how the VGAE is trained. The encoder involves two two-layered GCNs for obtaining $\mu$ and $\sigma$, each of which is realized with hidden channels equal to 256 and output channels equal to 128, while the two GCNs share the first layer. The dataset split process is the same as the training of the target GNNs. We train the VGAE model on the train set with the number of epochs equal to 100, batch size equal to 256, and learning rate equal to 0.001. The model that reaches the lowest loss on the validation set is saved for later OOD reweighting.

## B.2    More Quantitative Results

Here, we sequentially present the experimental results of 1) comparison between GEF [49] and our metrics *w.r.t* the latest dataset SHAPEGGEN and the other datasets (Table 3), 2) correlation between metrics and Recall across explanatory subgraphs in four node classification dataset (Table 4), and 3) correlation between metrics and Recall while evaluating the explanations generated by the inherent explainable GNN, GSAT [15] via introducing an additionally well-trained GIN [67] as the model $f$ in Algorithm 1 (Table 5).

Note that while evaluating explanations in node classification tasks, for each node in the input graph, we construct an ego graph for it based on the number of layers in the baseline GNN. Then, the explanation task for node classification can be transferred to the explanation task for graph classification. Furthermore, the remaining hyperparameters and methods in our OAR/SimOAR remain unchanged.

---

[2]Source code of DSE: `https://anonymous.4open.science/r/DSE-24BC/`

Table 5: Correlation between metrics and Recall while evaluating the explanations generated by the inherent explainable GNN, GSAT (corresponding to Figure 3 (a) in the main paper).

|        | BA3   | MUTAG | TR3   | MINST |
|--------|-------|-------|-------|-------|
| RM     | 0.376 | 0.381 | 0.402 | 0.337 |
| DSE    | 0.411 | 0.425 | 0.417 | 0.319 |
| OAR    | **0.613** | **0.590** | **0.585** | **0.606** |
| SimOAR | 0.598 | 0.572 | 0.552 | 0.581 |

## C  User Study

In order to measure the consistency between evaluation results and human intuition, we organized a large-scale user study engaging 100 volunteers. Each volunteer was asked to check 5 groups of graphs, which contain an instance from MNIST-sp, its predicted class, and 5 randomly sampled subgraphs from this instance, and try to rank these 5 subgraphs according to how well they serve as explanations of the prediction based on intuition. We exhibit partial results in Figure 5. According to these results we can find that our evaluation methods *i.e.* OAR and SimOAR show the highest consistency with human intuition.

