# OpenReview forum: "Evaluating Post-hoc Explanations for Graph Neural Networks via Robustness Analysis"
_NeurIPS.cc/2023/Conference — NeurIPS 2023 oral_

### Official Review · Reviewer_vGHJ · 2023-06-23

**Soundness:** 3 good
**Presentation:** 4 excellent
**Contribution:** 3 good
**Rating:** 7
**Confidence:** 4

**Summary:**

This paper presents a new metric, OOD-resistant Adversarial Robustness (OAR), for evaluating post-hoc explanation methods for Graph Neural Networks (GNNs). Inspired by adversarial robustness, OAR performs random perturbations on the complementary part of the explanation result, reducing the impact of the Out-Of-Distribution (OOD) issue common in previous removal-based metrics and ensuring consistency with GNNs' behavior compared to generation-based methods. The authors further introduce an OOD reweighting block, which measures the OOD score of each perturbed sample, allowing for the marginalization of OOD instances, and the OOD score is assigned via the reconstruction loss of a Variational Graph Auto-Encoder (VGAE) pre-trained by each dataset, inspired by graph anomaly detection methods. Extensive evaluations and ablation studies demonstrate that OAR aligns more closely with the ground truth.









**Strengths:**

- This work summarizes and addresses a critical and emerging issue in post-hoc explanation methods for GNNs, and the proposed metric is intuitive and can mitigate issues that may happen in previous removal-based and generation-based methods, offering significant benefits to the research community.
- The authors conduct extensive experiments, demonstrating the potential of the proposed metric as a superior evaluation tool for post-hoc explanation methods.
- It is interesting to see OOD score can be measured using graph anomaly detection methods (and that works).
- This paper is well-written and well-motivated.

**Weaknesses:**

- As an evaluation metric, OAR introduces an additional component that requires training on the original dataset used for the GNN. This process makes things complex, time-consuming, and may accumulate errors, making it tricky for different research groups to compare methods across various datasets.

- SimOAR somewhat addresses the above concern, but it does so at the expense of performance. From current experiments (w/ six backbone methods), the performance degradation seems to be significant.

- Though I am generally satisfied with evaluating six backbone explanation methods, the inclusion of more baselines would provide a more comprehensive statistical overview, given that this paper aims to introduce a metric for future community use. It would also be beneficial to include different types of post-hoc methods, such as decomposition, surrogate, and generation-based methods, among others.

**Questions:**

- I am not quite sure if the usage of "adversarial" and "attack" in OAR would be clear. I thought there would be some adversarial training in OAR but in fact the authors just got inspiration from them and OAR does not really do adversarial things.

- How is the metric RM calculated mathematically? Is it fidelity? Can the authors provide its formulation?

**Limitations:**

- This paper focuses on the explanation of graph-level tasks. How would this generalize to other types of tasks as a metric?
- There are works showing post-hoc explanation methods are always suboptimal in terms of finding label-relevant patterns and therefore proposing self-explainable models and pretrain-finetuning framework, e.g., GSAT. This may limit the future impact of OAR if it is only for post-hoc methods. Can the idea in OAR help self-explainable methods?

---

> ### Author Rebuttal · Authors · 2023-08-08
>
> *Dear Reviewer vGHJ:*
>
> *Thank you for the thoughtful feedback! Your constructive criticism has been invaluable in refining our work. Below, we give point-by-point responses. Hope our responses could address your concern!*
>
> >*W1 & W2: Concern about the training process in OAR, while SimOAR addresses the above concern but degrades the performance.*
>
> Thanks for your concern. We agree that the training process of OAR is complex and time-consuming. In fact, we have delved into this issue in **Appendix F** (Line 175-185) of the main paper. Specifically,
>
> 1. We believe that  learning distribution information from this training process  is inevitable to address the problem of OOD. Here is the reason:
>
>     - Explanation (which is indeed the process of sampling subgraphs from inputs) would inevitably introduce the risk of OOD. While to measure this OOD, it is necessary to obtain the distribution information.
>
> 2. Although SimOAR is a compromise which degrades the performance, we believe that the experimental results of OAR and SimOAR precisely demonstrate the potential optimization space. Concretely, **OAR and SimOAR represents two extremes:** OAR accurately acquires distribution to combat OOD, while SimOAR completely abandons this acquisition to improve efficiency. It naturally inspired the future optimized directions:
>     - On the one hand, we can find a balance between accuracy (OAR) and speed (SimOAR) to adapt to different tasks.
>     - On the other hand, we can explore more efficient generative algorithms that require less acquisition of distribution information, such as diffusion models, to optimize both accuracy and speed simultaneously.
>
> >*W2: It would also be beneficial to include different types of post-hoc methods.*
>
> Thanks for your suggestion. Following your guidance, we have conduct additional experiments across three types of post-hoc methods to validate the effectiveness of our OAR. Limited by the short rebuttal time, we only select one method for each type (*i.e.*, decomposition: **GNN-LRP**, surrogate: **PGMExplainer**, generation: **XGNN**) on BA3 and MNIST-sp. Moreover, we noticed that you are interested in **GSAT** in the following comments, hence, we also conduct the **GSAT** (in its post-hoc mode).
>
> Here are the results (corresponding to Figure 3 in the main paper):
>
> |||BA3|||
> |:-:|:-:|:-:|:-:|:-:|
> ||PGMExplainer|GNN-LRP|XGNN|GSAT|
> |RM|0.341|0.355|0.307|0.386|
> |DSE|0.409|0.412|0.367|0.370|
> |OAR|**0.488**|**0.525**|**0.505**|**0.463**|
> |SimOAR|0.463|0.500|0.482|0.432|
> ||||||
>
> |||MNIST-sp|||
> |:-:|:-:|:-:|:-:|:-:|
> ||PGMExplainer|GNN-LRP|XGNN|GSAT|
> |RM|0.302|0.323|0.287|0.342|
> |DSE|0.314|0.340|0.341|0.317|
> |OAR|**0.546**|**0.583**|**0.520**|**0.554**|
> |SimOAR|0.542|0.548|0.499|0.530|
> ||||||
>
> We will continue to conduct more complete experiments across other post-hoc methods and add them into the reversion.
>
> >*Q1: Concern about the usage of  "adversarial" and "attack".*
>
> We are sorry for making you confused. Specifically, our adversarial robustness mean that “How adversarial are random perturbations against the original data distribution, which are measured by OOD score”?
>
> Here are two more reasons why we named our attack as **adversarial attack**:
>
> - First, the starting point of our methods is finding the minimum perturbation leading to the wrong prediction (Line 128-132 in the main paper). However, this target is proved to be hard to reach and sometimes even intractable in the scenario given in the paper (Line 144-151). Thus, we transfer to approximate it from its dual perspective. That is, calculating the largest changes of outputs under the attacks. Hence, **our approach is essentially derived from the adversarial attack.**
>
> - Moreover, here are the imposed objective and the target of the *adversarial attack* and *our attack in OAR*, which are similar with each other:
>
> ||Objective|Target|
> |:-:|:-:|:-:|
> |Adversarial Attack|**selected part of input** (*which is selected by the attack algorithm*)|**perturb output as much as possible** (*until flip the outputs*)|
> |Attack in Our OAR|**selected part of input** (*which is selected by the explanations*)|**perturb output as much as possible** (*to see the robustness*)|
> ||||
>
> >*Q2: How is the metric RM calculated mathematically?*
>
> In our paper, Removal-based metric (RM) is employed to quantify the fidelity of the explanations. More formally, for an input graph $G$ and an explanation (subgraph) $G_s$, RM believes that a good explanation should have a large $f(G)-f(G_s)$, where $f$ is the GNN.
>
> >*L1: How would OAR generalize to other tasks?*
>
> Thanks for your concern. Following your comments, we have conducted experiments across several prevalent **node classification** datasets following [1].
>
> Here are the results:
>
> ||BA-Shapes|BA-Community|Tree-Cycles|Tree-Grid|
> |:-:|:-:|:-:|:-:|:-:|
> |RM|0.312|0.321|0.295|0.411|
> |DSE|0.406|0.377|0.343|0.384|
> |OAR|**0.542**|**0.560**|**0.489**|**0.440**|
> |SimOAR|0.527|0.535|0.471|0.431|
> ||||||
>
> According to these results, we can find that our OAR and SimOAR can work well for the task of node classification.
>
> *[1] GNNExplainer: Generating Explanations for Graph Neural Networks. NIPS 2019*
>
> >*L2: Can the idea in OAR help self-explainable methods?*
>
> Thanks for your concern. Yes, it can. The evaluation target of OAR is the explanation (subgraph). Hence, any algorithm which generate explanations can be evaluated by OAR.
>
> Following your concern, we have conducted the experiments on GSAT:
>
> |||GSAT|||
> |:-:|:-:|:-:|:-:|:-:|
> || BA3 | MUTAG | TR3 |MINST|
> |RM|0.376|0.381|0.402|0.337|
> |DSE|0.411|0.425|0.417|0.319|
> |OAR|**0.613**|**0.590**|**0.585**|**0.606**|
> |SimOAR|0.598|0.572|0.552|0.581|
> ||||||
>
> These results validate that our OAR can help self-explainable methods.
>
> *Once again, we sincerely appreciate your time and effort in reviewing our paper. Your criticism has been invaluable in refining our work, and we are more than happy to add clarifications to address your concerns!*
>
> *Best,*
>
> *Authors*

---

> > ### Comment · Reviewer_vGHJ · 2023-08-17
> >
> > Thank you for your detailed response and the additional experiments. These have addressed the majority of my questions and concerns. After reading the other reviews, I am pleased to maintain my original score. I appreciate the effort and insights the authors put into the paper.

---

### Official Review · Reviewer_uvv7 · 2023-06-30

**Soundness:** 4 excellent
**Presentation:** 3 good
**Contribution:** 4 excellent
**Rating:** 7
**Confidence:** 5

**Summary:**

The paper proposes a novel evaluation metric called OOD-resistant Adversarial Robustness (OAR) to address the issue of assessing the credibility of explanations provided by graph neural networks (GNNs). The paper criticizes existing evaluation methods that fail to consider out-of-distribution (OOD) data and can produce inconsistent results. OAR overcomes these limitations by calculating the robustness of explanatory subgraphs under attack and incorporating an OOD reweighting block. The paper also introduces a simplified version of OAR called SimOAR for more efficient evaluation of large datasets. Experimental results show that OAR outperforms current evaluation metrics and demonstrates consistency with metrics like Precision, Recall, and Human Supervision .

**Strengths:**

1. This work tackles an important research gap on evaluating the explanation for GNN.
2. The paper is well-written and easy to follow.
2. The authors conduct experiments, and the results demonstrate the advantage of their proposed method.

**Weaknesses:**

1. Figure 1 is kind of hard to follow. It could be better if authors can refine it.
2. Some experiment setting is missing. For example, what is the setting of the GAE? What is the number of the generated graph?
3. Lack of related work, and some important reference is missing. For example, the authors could add a section introduce the explanation methods for GNN according to [1].

[1]  Zhang, He, et al. "Trustworthy graph neural networks: Aspects, methods and trends." arXiv preprint arXiv:2205.07424 (2022).

**Questions:**

Listed in Weakness.

**Limitations:**

Yes.

---

> ### Author Rebuttal · Authors · 2023-08-08
>
> *Dear Reviewer uvv7:*
>
> *We gratefully thank you for your valuable comments! Here we meticulously give point-by-point responses to your comments. Hope that our responses could address your concerns!*
>
> >**W1: Figure 1 is kind of hard to follow. It could be better if authors can refine it.**
>
> Thanks for your concern. Following your suggestion, we have made significant refinements to improve its clarity. The refined version provides a more coherent representation of the information it conveys, as shown in the new added *Supplementary PDF*.
>
> The following are some specific modifications:
>
> - We have extracted the common parts of Figure 1 (a), Figure 1 (b), and Figure 1 (c), and placed them together to avoid repetition. We have also added appropriate textual descriptions to make them easier to understand.
>
> - We have increased the font size to match the text in the main body.
>
> - We have provided detailed illustrations of Figure C (our algorithm OAR) to demonstrate the process of our algorithm more intuitively and smoothly.
>
> We hope that the revised Figure 1 now effectively supports the content and contributes to the comprehension of our research.
>
> Thanks again for your valuable suggestion!
>
> >**W2: Some experiment setting is missing. For example, what is the setting of the GAE? What is the number of the generated graph?**
>
> Thanks for your concern. For the setting of the GAE, please refer to **Line 138-140** in Appendix C;  for the number of the generated graph, please refer to **Line 160-161** in Appendix C, where we point out that for BA3, TR3, and MUTAG, the number of perturbed subgraphs $N_{perturb}$ is 20, while for MNIST-sp, the number of perturbed subgraphs $N_{perturb}$ is 50 owing to its large size.
>
> We agree with your concern about the absence of these important parameters in the main paper. Hence, following your suggestion, **we have moved them from the Appendix C to the Experimental Setup** section in the main paper of the revised version. We hope that this modification could enhance the readability and reproducibility of our paper.
>
> >**W3: Lack of related work, and some important reference is missing. For example, the authors could add a section introduce the explanation methods for GNN according to [1].**
>
> Thanks for your concern. Based on your recommendation, we have included the missing reference [1] within the Related work section in the main paper, ensuring proper attribution to the original work.
>
> Furthermore, following your suggestion, we have introduced the current trustworthy GNNs from six aspects (robustness, explainability, privacy, fairness, accountability, and environmental well-being) in the Related Work following [1].
>
> By incorporating these suggested changes, we believe that we have enriched the manuscript and provided readers with a more comprehensive understanding of the explanation methods for GNN.
>
> [1] Zhang, He, et al. "Trustworthy graph neural networks: Aspects, methods and trends." arXiv preprint arXiv:2205.07424 (2022).
>
> *Thank you for drawing our attention to these important aspects. Your valuable feedback has greatly enhanced the quality of our research. If you have any further **recommendations** or require additional **clarification**, please do not hesitate to let us know.*
>
> *Best,*
>
> *Authors*

---

> > ### Comment · Reviewer_uvv7 · 2023-08-18
> >
> > I appreciate the comprehensive reply and the extra experiments you conducted. They have resolved most of my inquiries and worries. Having gone through the other reviews, I'm content to keep my initial rating.

---

### Official Review · Reviewer_xH8P · 2023-07-06

**Soundness:** 3 good
**Presentation:** 3 good
**Contribution:** 3 good
**Rating:** 6
**Confidence:** 4

**Summary:**

This paper presents a new evaluation metric OAR for evaluating post-hoc explanation methods for GNNs. OAR has two main advantages in that it does not need ground-truth labels and in that it prevents finding spurious explanations based on out-of-data-distribution phenomena. The paper focuses on graph classification: The input for this explanation is a subgraph of the graph for which we want to explain a prediction. The proposed method OAR, creates several permutation of the original graphs, where it is allowed to only permute edges **not** in the explanation. A generative model forces those permutations to not leave the training data distribution. The less the permutations can change the model prediction the better an explanation we consider the subgraph. Experiments validate the effectiveness of OAR.


**Strengths:**

The paper tackles an important topic, GNN explanation, in an ambitious way: To evaluate different explanation methods unsupervised, without access to ground truth explanations.

The paper is in a good shape (apart from the introduction+related work): The writing is clear and allows for understanding most of the paper in the first or second read. The figures are helpful for understanding and well connected to the text. The experiments chosen by the authors make sense and I appreciate the authors also running a user study.


**Weaknesses:**

I do not like the approach the authors have taken for the related work in this paper. The first section is a hybrid of related work and introduction, and I feel it serves neither well. I am missing a concise motivation and a broader context of the paper from this section. The actual related work is in Appendix A, which I would prefer to be in the main paper. I also think that there is an important branch of work missing [1-5]. These works discuss general properties that we may want an explanation method and an evaluation setup to adhere to. I think it would benefit the paper to discuss how OAR is compatible/incompatible with these proposals. These works also offer some alternative benchmarks for evaluation instead of the flawed BA and tree-based (TR) datasets.
[1] Himmelhuber et al: Demystifying Graph Neural Network Explanations
[2] Sanchez-Lengeling et al. Evaluating attribution for graph neural networks
[3] Agarwal et al. Evaluating explainability for graph neural networks
[4] Agarwal et al. Probing GNN explainers: A rigorous theoretical and empirical analysis of GNN explanation methods
[5] Faber et al. When comparing to ground truth is wrong: On evaluating GNN explanation methods

There is also one paper discussing out-of-distribution versus explanation for GNNs [6] that might be worth discussing/comparing to.
[6] Faber et al. Contrastive graph neural network explanation

I am not convinced about the explanations that the method produces. Generally, we want to create explanations to make humans understand what is happening. For this, every step to create the explanation is ideally human inspectable. Here, it seems like we are transferring the trust problem from the GNN to the VGAE. This becomes the blackbox that somehow determines if the results are good or bad and we cannot inspect this blackbox. For example, using recall on ground-truth data makes for a good explanation metric because us humans can see *why* the explanation is supposedly good.


**Questions:**

Could you expand on the setup used for the user study? The previous years (e.g., https://neurips.cc/Conferences/2021/PaperInformation/PaperChecklist) included a paper checklist of what to consider and the details in Appendix D are quite short.

The study on MNIST explanations seems to be very dependent on the layout of the explanation subgraph (maybe even more so than the choice of what nodes to put there). Can you expand on how you created the graph layout? Are superpixel nodes positioned where they are placed in the image?

Does the permutation space for the targeted adversarial attacks allow adding and deleting nodes? Or are the attacks constrained to happen in the given adjacency matrix?

OAR itself makes no assumptions on the size of the explanation, but it seems it is easier to get good explanations the larger the explanation graph is: there is less attack space for adversarial permutation and a larger fixed graph likely helps for low OOD scores. I wonder if OAR would benefit from a component that penalizes surplus nodes in the explanation?

Could the method be expanded to also support node classification? One naive angle might be to extract the receptive field of each node and use OAR on those graphs?

From what I understand, an important motivation is the fight of out-of-distribution data. This motivates the generative graph model to find and remove outliers. On the other hand, it seems that SimOAR performs similarly without using such a model but a budget(?) on perturbations. Can we maybe do something simpler than the generative model, for example, looking at spectra?

Some of the works linked above showed that the BA and tree based datasets have flaws and GNNs do not always use all the data present in the ground truth. Could these also influence your experiments?


**Limitations:**

Addressed by the authors

---

> ### Author Rebuttal · Authors · 2023-08-07
>
> Thank you very much for your valuable comments. We hope that the following reply could address your concerns!
>
>
> >**W1: Concerns about the Introduction and the missing related work [1-5].**
>
> Thanks for bringing this point to us. Following your suggestions, we have implemented the following changes in the response and our revision:
>
> 1. **Introduction & Related Work Organization**. The 'Introduction' section has been revamped to streamline previous studies. The 'Related Work' section has been added into the main paper.
>
> 2. **Literature View**. We appreciate your introduction to references [1-5] and have incorporated them into the 'Related Work' section. These papers primarily classify current metrics into four categories: accuracy, faithfulness, stability, and fairness. Notably, OAR align with the 'faithfulness' metric.
>
> 3. **Evaluation Metric & Dataset**. Drawing from references [1-5], we've specifically adopted the contemporary faithfulness metric, GEF [3], and the latest synthetic dataset, SHAPEGGEN [3], to enhance our paper. We utilized these resources to perform additional experiments, with the outcomes presented in the table below and Table 1 of the main paper.
>
> ||SHAPEGGEN|TR3|MUTAG|MNIST|BA3|
> |:-:|:-:|:-:|:-:|:-:|:-:|
> |GEF|0.800|0.734|0.800|0.934|0.734|
> |SimOAR|0.800|0.867|0.800|0.934|0.934|
> |OAR|0.867|0.934|1.000|0.934|1.000|
> |||||
>
> > **W2: [6] might be worth discussing/comparing to.**
>
> Thanks for your suggestion. We have incorporated a discussion on CoGE [6] in our revision.
>
> In essence, although CoGE can determine if a subgraph is associated with the OOD class, it fails to provide a precise score delineating the OOD degree, making it coarser compared to OAR.
>
> Moreover, CoGE is more aligned with explanation methodologies rather than functioning as an evaluation framework, which is why we did not select it as a baseline.
>
> We hope this classification addresses your concerns!
>
> > **W3: Concerns about transferring the trust problem from the GNN to the VGAE.**
>
> Thank you for highlighting this important aspect. While we recognize the concerns of transferring trust to VGAE with OAR, we assert that its impact is relatively minimal for several reasons:
>
> 1. **VGAE-guided OAR**. Indeed, the trust and accuracy of VGAE is important to evaluate the explanation faithfully. However, we are optimistic that as generative research (e.g., diffusion models) advances, emerging generation ability will mitigate this limitation.
>
> 2. **VGAE-free SimOAR**. Our streamlined variant, SimOAR, sidesteps VGAE altogether. Instead, it utilizes transparent heuristics for perturbation generation. This strategic choice inherently counteracts the transfer of trust issues from GNN to VGAE.
>
> 3. **Performance Insights**. As Table 1 shows, SimOAR is slightly worse or on par with OAR, but outperforms all baselines. This underscores the idea that VGAE's influence is minimal.
>
> We're grateful for your insightful suggestions, and we will incorporate them into the reversion.
>
> >**Q1: Expand the setting of user study following checklist.**
>
> Thanks for your suggestions. We have detailed the setting following checklist and incorporated it into our revision:
>
> 1. **Instructions.** Each participant was asked to check 5 groups of graphs as shown in Appendix D and answer which subgraph is best preserves digital information.
> 2. **Risks.**  Our user study do not have any risks.
> 3. **Wage.** The participants in our user study were volunteers.
>
> >**Q2: How to creat the MNIST-sp?**
>
> We created the MNIST-sp according to [1]. Specifically, the node features are the pixels' values and the  pixels' centers. Edges are the spatial distance between the superpixel centers.
>
> *[1] Geometric deep learning on graphs and manifolds using mixture model cnns. CVPR. 2017*
>
> >**Q3: Does the attacks allow adding and deleting nodes?**
>
> No, our attacks are only allowed to happen in the given adjacency matrix. We will explore the addition/deletion of nodes in future work.
>
> >**Q4: If OAR would benefit from penalizing surplus nodes.**
>
> Following your recommendations, we integrated the L1-norm for explanation size control. Experiments exhibted in the following and the Table 1 of the main paper, verify that the Component you suggested indeed enhances our methods.
>
> ||TR3|MUTAG|MNIST|BA3|
> |:-:|:-:|:-:|:-:|:-|
> |SimOAR|0.867|0.800|0.934|0.934|
> |SimOAR+L1|**0.934**$\uparrow$|0.800|0.934|**1.000**$\uparrow$|
> |OAR|0.934|1.000|0.934|1.000|
> |OAR+L1|0.934|1.000|**1.000**$\uparrow$|1.000|
> |||||
>
> >**Q5: Could OAR work for node classification?**
>
> Yes, it can. Here are the results on four node classification datasets following [1]:
>
> ||BA-Shapes|BA-Community|Tree-Cycles|Tree-Grid|
> |:-:|:-:|:-:|:-:|:-:|
> |RM|0.312|0.321|0.295|0.411|
> |DSE|0.406|0.377|0.343|0.384|
> |OAR|**0.542**|**0.560**|**0.489**|**0.440**|
> |SimOAR|0.527|0.535|0.471|0.431|
> |||||
>
> *[1] GNNExplainer: Generating Explanations for Graph Neural Networks. NIPS 2019*
>
> >**Q6: Can we do something simpler than VGAE, like spectra?**
>
> Yes, we can. Spectra is the commonly used techniques in anomaly detection. Considering their typically work for nodes classification, we use the Manhattan distance between the spectrum as the OOD degree and exhibit the results:
>
> ||MUTAG|BA3|TR3|MNIST|
> |:-:|:-:|:-:|:-:|:-:|
> |OAR|0.567|0.503|0.483|0.455|
> |SimOAR+Spectra|0.511|0.459|0.449|0.433|
> |||||
>
> Based on these we find that the performance of spectra is inferior to that of VGAE.
>
> >**Q7: Could BA3 and TR3 influence experiments?**
>
> Thanks for your concern. We believe that they will not. Here are two reasons:
>
> - First, although ground truth in BA3 and TR3 might not conform to the decision-making process exactly, they contain sufficient discriminative information to help justify the quality of explanations.
>
> - Second, even after excluding the potentially problematic datasets BA3 and TR3, we still have other datasets like MUTAG and MNIST-sp, and our methods also achieved the best performance on these datasets across various settings.

---

> > ### Comment · Reviewer_xH8P · 2023-08-15
> >
> > Thank you for your very detailed review and in particular for the additional experiments.
> >
> > #### W1+W2
> > Thank you for rearranging the paper structure and expanding the related work section. In particular, I appreciate that the authors extend this discussion into new experimental results. The results provide further support for their proposed method. I agree that [6] can just be discussed and does not warrant experiments as a baseline.
> >
> > #### Q1, Q2, Q3, Q5, Q7
> > Thank you for providing additional details for these questions. I also appreciate here taking the extra effort to create the additional node classification experiment. Can you describe a bit how you adapted OAR to the node setting?
> >
> > #### Q4
> > Thank you for taking the time to also try this experiment.
> >
> > #### W3, Q6
> > While I am not fully convinced by the argument, I agree with SimOAR as an inspectable alternative to the full VGAE-OAR version. The reason why I am saying I am not fully convinced is that there are differences in performance, even when we help SimOAR with spectral information (from the experiment in Q6).
> >
> > Overall, I really like the new information from this rebuttal phase. The authors provided four new sets of experiments. They embedded current GNN explanation work in their framework, extended to Node classification, found improvements with L1 regularization and furher compared to a spectral baseline. My assessment of losing inspectability was too pessimistic because of SimOAR, although it lacks slightly behind in performance. Therefore, I increase my score.

---

> > > ### Author Response · Authors · 2023-08-17
> > > **Response to Reviewer xH8P**
> > >
> > > Dear Reviewer xH8P,
> > >
> > > I would like to express my sincere gratitude to you for recognizing and providing constructive feedback on our work, which has been invaluable in refining it.
> > >
> > > Following your comments regarding node classification, we have incorporated the experimental details in the revision. Specifically, for each node in the input graph, **we construct an ego graph for it based on the number of layers in the baseline GNN.** Then, the explanation task for node classification can be transferred to the explanation task for graph classification. Furthermore, the remaining hyperparameters and methods in our OAR/SimOAR remain unchanged.
> > >
> > > Once again, I would like to extend my sincere appreciation for the time and effort you have dedicated to the review process! We truly value the importance of your input in our professional journey!
> > >
> > > Best regards,
> > >
> > > Authors

---

### Official Review · Reviewer_QMrv · 2023-07-07

**Soundness:** 3 good
**Presentation:** 3 good
**Contribution:** 4 excellent
**Rating:** 8
**Confidence:** 4

**Summary:**

This paper studies the explainability evaluation of GNNs, and proposes a new evaluation paradigm. Inspired by adversarial robustness, it uses a generative model (VGAE) to fulfill the explanation subgraph, randomly perturbs the fulfilled parts, and uses the OOD scores of perturbed graphs to measure the importance of explanation subgraph. Experiments are done on several datasets and show the improved evaluation quality w.r.t. diverse criteria.

**Strengths:**

1.	This paper summarizes the removal- and generation-based evaluation protocols well, and points out their drawbacks w.r.t. post-hoc explainability. It resolves these drawbacks by calculating the robustness of explanations under attack and OOD-reweighting. This motivation is clear and reasonable.
2.	In terms of technical contributions, the evaluation of explanation robustness is well-designed. It first gives an adversarial robustness-related definition, and then transfers it into a tractable objective and designs an OOD-reweighting block (i.e., an external VGAE) to solve the objective. Moreover, a computationally efficient variant is also proposed.
3.	The experiments are sufficient to demonstrate the effectiveness of the proposed method, w.r.t., explanation evaluation, generalization, model design, and user study.
4.	I appreciate the diverse criteria used in the paper, especially the “consistency of ground-truth explanations and human intuition”.
5.	The presentation of the proposed method is clear.


**Weaknesses:**

1.	Regarding adversarial robustness, the proposed measurement is based on perturbing the subgraphs. I have two questions: (1) why name the random perturbation as adversarial robustness, which usually performs adversarially perturbation? Does it mean that “How adversarial are random perturbations against the original data distribution, which are measured by OOD score”? (2) How many perturbed subgraphs are needed? I think these concepts are essential to understand the proposed evaluation method. Hence, more clarification is needed.
2.	The OOD reweighting block is implemented by VGAE, hence, the proposed method seems heavily dependent on the VGAE quality. However, many studies show the generative ability of VGAE is suboptimal and degenerated. It would be better to analyze the influence of VGAE quality on the OAR performance.


**Questions:**

1.	Why name the random perturbation as adversarial robustness, which usually performs adversarially perturbation? Does it mean that “How adversarial are random perturbations against the original data distribution, which are measured by OOD score”?
2.	How many perturbed subgraphs are needed?
3.	What is the influence of VGAE quality on the OAR performance?

---

> ### Author Rebuttal · Authors · 2023-08-08
>
> **Dear Reviewer QMrv:**
>
> **Thank you for the thoughtful feedback! Your constructive criticism has been invaluable in refining our work. Below, we give point-by-point responses to your comments. Hope that our responses could address all your concerns!**
>
>
> >*W1 & Q1: Does the adversarial robustness mean that “How adversarial are random perturbations against the original data distribution, which are measured by OOD score”?*
>
> Thanks for your concern. Yes, it does.
>
> Here are two more reasons why we named our methods OAR (OOD-resistant **adversarial** robustness):
>
> - First, the starting point of our methods is finding the minimum perturbation leading to the wrong prediction (Line 128-132 in the main paper). However, this target is proved to be hard to reach and sometimes even intractable in the scenario given in the paper (Line 144-151). Thus, we transfer to approximate it from its dual perspective. That is, calculating the largest changes of outputs under the attacks. Hence, **our approach is essentially derived from the adversarial attack.**
>
> - Moreover, here are the imposed objective and the target of the *adversarial attack* and *our attack in OAR*, which are similar with each other:
>
> |               |    &emsp; &emsp;&emsp; &emsp; &emsp; &emsp; Objective  |   &emsp; &emsp;&emsp; &emsp; &emsp; &emsp;Target  |
> |:-------------:|:--------:|:--------:|
> |     Adversarial Attack    |   **selected part of input** (*which is selected by the attack algorithm*)   |   **perturb output as much as possible** (*until flip the outputs*)    |
> |     Attack in Our OAR   |   **selected part of input** (*which is selected by the explanations*)  |   **perturb output as much as possible** (*to see the robustness*)   |
> ||||
>
> Moreover, following your suggestion, we have added these clarifications into the Introduce Section in the reversion. We believe that this will greatly increase the readability, coherence, and comprehensibility of the article.
>
> Thanks again for your valuable suggestion!
>
> >*W2 & Q2: How many perturbed subgraphs are needed?*
>
> Please refer to Line 160-161 in Appendix, where we point out that for BA3, TR3, and MUTAG, the number of perturbed subgraphs $N_{perturb}$ is 20, while for MNIST-sp, the number of perturbed subgraphs $N_{perturb}$ is 50 owing to its large size.
>
> >*W3 & Q3: What is the influence of VGAE quality on the OAR performance?*
>
> Thanks for your concern. The quality of VGAE will have a certain impact on the performance of OAR, but it is not significant. Here is the reason:
>
> - The simplified version of OAR -- SimOAR (which can be viewed as the case that the VGAE is poorly trained and produces random scores) -- performs close to OAR. It indicates that the performance of VGAE does not significantly impact OAR's performance, and further demonstrates that the effectiveness of our OAR and SimOAR is mainly attributed to the preferable paradigm, instead of the other module (e.g., VGAE).
>
> **Once again, we sincerely appreciate your time and effort in reviewing our paper. Your constructive criticism has been invaluable in refining our work, and we are more than happy to add clarifications to address any additional recommendations and reviews from you!**
>
> **Best,**
>
> **Authors**

---

> > ### Comment · Reviewer_QMrv · 2023-08-15
> > **Response to Authors and raise score from 7 to 8**
> >
> > Thank you for answering my comments. Your response address the concerns I had, and I will raise my score.

---

> > > ### Author Response · Authors · 2023-08-15
> > > **Response to Reviewer QMrv**
> > >
> > > Dear Reviewer QMrv,
> > >
> > > I would like to express my heartfelt gratitude to you for recognizing our work. Your insightful guidance and constructive suggestions have undoubtedly played a vital role in improving the quality of our work.
> > >
> > > I would also like to extend my sincere appreciation for the time and effort you have dedicated to the review process. We truly value the importance of your input in our professional journey!
> > >
> > > With warm regards,
> > >
> > > Authors

---

### Author Rebuttal · Authors · 2023-08-10

Dear Reviewers:

We gratefully thank you for your valuable comments! We were encouraged to hear that our work has **clear and well-written presentations** (by all Reviewers), **well-designed and interesting technical contributions** (by Reviewer QMrv and vGHJ), **extensive and sufficient experiments** (by all Reviewers), which **addresses a critical and emerging issue** to the research community (by Reviewer uvv7 and vGHJ).

Here we meticulously give point-by-point responses to your comments, and further add the additional experiments and figures into the one-page supplementary PDF. Especially, we have taken measures to enhance the structure of the Introduction and Related Work section, and provided a more rigorous definition of our methods. Furthermore, we have provided a more detailed description of our experimental settings and included a wider range of representative baselines and datasets for conducting additional experiments. We hope that our responses adequately address all your concerns and meet the expectations of the conference committee.

Once again, we sincerely appreciate your time and effort in reviewing our paper. Your constructive criticism has been invaluable in refining our work, and we are more than happy to add clarifications to address any additional recommendations and reviews from you!

Best,

Authors

---

### Decision · Program_Chairs · 2023-09-21

**Decision:**

Accept (oral)

**Comment:**

The reviewers are all excited about this work and unanimously recommended acceptance.

Among all strengths that reviewers mentioned, this paper is particularly well motivated and focuses on a very important problem, tackling a gap between on evaluating the explanation for GNN. The detailed discussion on the drawbacks of different evaluation protocols is also appreciated. The experimental study is solid, with user study and diverse criteria.